# Why Are Young People Willing to Pay for Health? Chained Mediation Effect of Negative Emotions and Information Seeking on Health Risk Perception and Health Consumption Behavior

**DOI:** 10.3390/bs14100879

**Published:** 2024-09-30

**Authors:** Jia Li, Yingyi Li

**Affiliations:** 1School of Journalism and Information Communication, Huazhong University of Science and Technology, Wuhan 430074, China; d201881179@hust.edu.cn; 2School of Journalism and Communication, Shandong University, Jinan 250199, China

**Keywords:** health consumption behavior, health risk perception, negative emotions, information seeking, chained mediation

## Abstract

The perception of health risks can influence people’s health behaviors. However, in the context of modern consumer society, few people delve into in-depth discussions on health consumption as a form of health protection behavior. Inspired by the Health Belief Model and Protection Motivation Theory, this study interprets health consumption behavior as a new form of health protection behavior. A survey was conducted on a sample of Chinese youth (N = 885) to explore the mechanisms of action between health risk perception and health consumption behavior using structural equation modeling. The study found that: (1) health risk perception has a significant positive impact on the health consumption behavior of young people; (2) negative emotions and information seeking play mediating roles respectively in the mechanism of the impact of health risk perception on health consumption behavior; and (3) in addition to their individual mediating roles, negative emotions and information seeking behavior collectively play a chained mediation role in this process. Implications of these results, both theoretical and practical, are further discussed.

## 1. Introduction

In recent years, with the increase in per capita disposable income, the aggravation of population aging, and the enhancement of health awareness, “health consumption” has gradually become a prevalent phenomenon in China. Data from the China Consumers Association reveal that there is a continuous increase in the demand among Chinese residents for health foods, food for special medical purpose, home health products, and precise nutrition supplements [1]. Moreover, there is a trend evolving from “selective consumption” towards “health essentials”. This suggests that in order to reduce the risk of illness and maintain personal health, individuals are not only adopting traditional health behaviors such as quitting smoking, moderating alcohol consumption, having a balanced diet, and engaging in moderate exercise, but also considering health consumption as an effective health behavior. Therefore, it is necessary to carefully consider the mechanisms and influencing factors of health consumption as a novel health behavior.

Health risk perception is an important variable in health behavior research. In the past few decades, researchers have extensively discussed the complex relationship between health risk perception and health behavior within theoretical frameworks such as the Health Belief Model and Protection Motivation Theory. Studies across various fields have also confirmed the significant impact of health risk perception on health behavior [2,3,4]. Moreover, studies have shown that health risk perception can influence individuals’ health consumption behavior. People are willing to increase their purchase intention for specific products in order to reduce the health risks they may face [5,6]. Additionally, interdisciplinary research has shown that negative emotion and information seeking are important factors influencing individual consumption behavior. On the one hand, studies in the fields of branding, marketing, and management have demonstrated that negative emotions can significantly impact consumer purchasing decisions [7,8]. For example, fear of death can heighten individuals’ psychological anxiety and feelings of insecurity, leading to a stronger inclination towards materialism [9] and the pursuit of materialistic buying behavior [10]. On the other hand, according to consumer decision-making process models, information seeking is a crucial step for individuals in their consumer behavior [9], and the increasingly complex information environment has made information seeking an indispensable “new step” in consumer behavior [11].

However, there is currently no research that conceptualizes health consumption behavior as a novel type of health behavior and considers negative emotion and information seeking as mediating variables to investigate the potential mechanisms underlying the relationship between health risk perception and health consumption behavior. In light of this gap, this study focuses on 885 Chinese youths and aims to develop a multilevel structural equation model. This model explores, for the first time, the potential impact of health risk perception on health consumption behavior through the mediating roles of negative emotion and information seeking. Through this endeavor, we aim to extend the theoretical boundaries of traditional health behavior research and enhance people’s understanding of health consumption behavior.

### 1.1. Health Consumption Behavior

Health behavior refers to the proactive actions individuals take to prevent diseases and maintain their well-being, primarily including healthy dietary habits, not smoking, regular exercise, and maintaining a healthy weight [12]. According to the Health Belief Model, the primary motivation for engaging in health behavior is to avoid negative health outcomes [13]. When faced with different perceptions of health risks, individuals exhibit proactive coping behaviors or avoidance behaviors to address potential hazardous situations [14]. Therefore, “health behaviors” essentially consist of tangible actions individuals take to reduce the health risks they face. Similarly, health consumption represents a form of purchasing behavior undertaken by individuals with the aim of reducing their health risks. By purchasing health foods, food for special medical purpose, home health devices (such as air purifiers, water purifiers), and fitness programs, individuals seek to reduce uncertainties about health and the associated harm. Thus, health consumption should be considered as a novel form of health behavior [12].

### 1.2. Health Risk Perception and Health Consumption Behavior

The concept of health risk perception originates from risk perception, referring to individuals’ attitudes and intuitive judgments regarding the likelihood of experiencing negative health outcomes from objective health risks [2]. It has been confirmed that health risk perception is highly related to individuals’ health behaviors [3,15]. A higher level of risk perception can enhance individuals’ willingness to quit smoking [16], get vaccinated [3], and engage in preventive health behaviors [17]. Furthermore, the influence of health risk perception on consumer behavior has also been confirmed. The perceived health risks and external threat-related health information individuals perceive in the social environment can affect their purchase decisions and behaviors [18,19]. Existing research indicates that individuals are more inclined to consume products and services that can help them reduce or alleviate health threats to cope with potential negative impacts [20]. For example, exposure to mortality threat information prompts individuals to be more willing to purchase green health foods [5].

According to the Health Belief Model, an increase in health risk perception will motivate individuals’ self-protective behaviors [13]. Therefore, the higher the level of health risk perception in individuals, the stronger their willingness to engage in self-protection and more likely they are to exhibit consumption behaviors aimed at reducing their own health risks. For example, individuals typically purchased masks or disinfectants during the COVID-19 pandemic to reduce the risk of infection. Therefore, the present study proposes the following hypothesis:

**H1:** 
*Health risk perception has a positive promoting effect on the health consumption behavior of young populations.*


### 1.3. Mediating Role of Negative Emotions

Emotion is the psychological and physiological state that individuals experience in response to specific stimuli [21] and is an important variable in risk perception and behavior research [22,23,24]. Studies have shown that risk perception can induce emotions [25], and individuals’ emotional responses are the result of their risk perception [26]. High levels of risk perception may lead individuals to experience negative emotions such as fear or worry [27,28].

On the other hand, it has been confirmed that negative emotions can influence individual behavior [29]. If individuals are triggered by negative emotions such as anger and sadness, it can affect their subsequent behavior [30,31]. Studies have shown that negative emotions can impact individuals’ purchasing behavior [32,33]. People can meet their specific needs through self-compensation mechanisms in their purchasing behavior and tend to release unsettling negative emotions by increasing their purchasing behavior [34].

In general, when individuals experience negative emotions such as anxiety, worry, and fear due to health risk perception, they tend to consider possible coping strategies [35]. If specific coping strategies are perceived as effective in alleviating the threat and easy to implement, it can generate a motivation to protect and trigger specific behaviors [36]. Since health consumption behavior not only effectively releases negative emotions but is also easier to adopt compared to traditional health behaviors like quitting smoking, drinking, or regular exercise, it may be perceived as a daily health behavior to be adopted by individuals. Therefore, based on the classification of emotional types by Watson and Tellegen [34], this study considers positive and negative emotions as two relatively independent fundamental dimensions and puts forward the following hypothesis:

**H2:** 
*Negative emotions, such as fear, anxiety, and worry, play a mediating role between health risk perception and health consumption behavior in the young population.*


### 1.4. Mediating Role of Information Seeking

In the study of health communication, health information seeking behavior (HISB) refers to individuals seeking information about health, risks, diseases, and health protection in specific events or situations [37]. Research indicates that the higher the individual’s perception of health risks, the stronger their motivation to acquire information [38], and a higher level of health risk perception can trigger more frequent health information seeking behavior [39,40].

Furthermore, the impact of information seeking on health behavior has been confirmed in both traditional and digital media contexts [41]. For instance, the more frequently people use traditional media such as radio and print, the stronger their willingness to receive vaccinations [42]; engaging in information seeking on digital media was a significant influencing factor for individuals to directly or indirectly adopt preventive behaviors during the COVID-19 pandemic [43].

Based on the above logic, a higher level of health risk perception will lead to more frequent information seeking, and the increase in information seeking will then stimulate individuals to generate or modify their own health behaviors. Although relevant studies have not explored the potential correlation between information seeking and health consumption behavior, a substantial amount of research on consumer behavior has demonstrated that information seeking not only alters individual purchasing behavior [44] but also positively impacts individual purchase decisions [45]. Therefore, this study considers that individuals’ information seeking is both a result of triggered health risk perception and a cause of health consumption behavior, and we propose the following hypothesis:

**H3:** 
*Information seeking plays a mediating role in the relationship between health risk perception and health consumption behavior in youths.*


### 1.5. The Chained Mediation Role of Negative Emotion and Information Seeking

As mentioned earlier, this study assumes that negative emotions and information seeking are important factors influencing the health consumption behavior of young people and that perceived health risks catalyze both of the above. Therefore, there could be a correlation between negative emotions and information seeking in the mechanism of health risk perception and health consumption behavior. However, there have been no studies discussing this issue to date.

Existing research has shown that individual emotional responses related to risk significantly influence the way they process subsequent risk information [46,47]. Griffin et al. [48] pointed out that under the dual drive of negative emotions such as worry and anger, and the principle of information sufficiency, individuals usually engage in more detailed information seeking to achieve their ideal level of information sufficiency. Based on the above research, the negative emotions generated by individuals’ perception of health risks may promote their information seeking and further stimulate health consumption behavior. In other words, individuals with negative emotions are more inclined to search for information compared to others. Therefore, this study suggests that the mediating effect of negative emotions and information seeking between health risk perception and health consumption behavior is not mutually independent; negative emotions serve as a precursor to information seeking, and both factors play a chained mediation role in the interactive mechanism of health risk perception and health consumption behavior. Therefore, this study proposes the following hypothesis:

**H4:** 
*Negative emotions and information seeking play a chained mediation role between health risk perception and health consumption behavior in young people.*


A summary of the hypotheses is described in Figure 1.

## 2. Materials and Methods

### 2.1. Sample

The data come from youths in China. They are mainly based on the following considerations: First, there’s a discernible trend among young individuals favoring health and wellness products over traditional health preservation activities. Second, this demographic exhibits a marked tendency to delve into health issues through online information seeking. Informed consent was sought from each participant prior to data collection, and all youth volunteered to participate in the questionnaire.

In order to test the above hypotheses, we conducted an online survey from January to April 2023. Through quota sampling, a total of 973 questionnaires were collected. After excluding the data with vacant response content, too short response time, and obvious repeated answers, 885 valid questionnaires were retained, with an effective rate of 91.0%. The age range of the respondents was 18 to 35 years, with a mean age of 26.97 ± 4.51 years. Among them, 409 (46.21%) were female, and 476 (53.79%) were male.

### 2.2. Measures

This study encompasses four primary variables alongside demographic factors. These comprise the independent variable “Perception of Health Risks,” mediating variables “Negative Emotions” and “Information Seeking”, and the dependent variable “Health Consumption Behavior”. Each variable was gauged by either referencing or adapting from established and validated scales, summing up to a total of 23 items. All items were assessed using a Likert five-point scale, where “1” signifies strong disagreement and “5” denotes strong agreement. Demographic variables controlled for in the analysis included gender, age, highest educational attainment, and major field of study.

#### 2.2.1. Health Risk Perception

Based on Ajzen and Madden’s [49] framework for assessing health risk perception, we gauged the youth demographic’s perception across two distinct dimensions: disease susceptibility and disease severity. The former captures an individual’s belief about the likelihood of experiencing a health risk event, while the latter delves into the anticipated impact and intensity of such events as perceived by the individual. Using a 5-point Likert scale (1 = strongly disagree, 5 = strongly agree), respondents’ health risk perceptions were determined through the following seven items: (1) Given my present circumstances, I am at risk of contracting a specific disease. (2) Illnesses can strike anyone at any moment. (3) There’s a possibility of future health adversities affecting me or a family member. (4) Presently, there is an escalating severity of health issues among people. (5) An illness afflicting a family member would have enduring consequences on my family. (6) Diseases invariably inflict physical discomfort. (7) A significant number of people face mortality due to illnesses.

For analysis, the questionnaire responses were averaged to formulate a health risk perception index. A higher score directly correlates with heightened health risk perception (M = 3.38, SD = 0.82, Cronbach’s α = 0.87).

#### 2.2.2. Negative Emotion

To measure negative emotions related to health and disease, we categorized negative emotions into three types: anxiety, worry, and fear. Drawing upon the research by Brug [50], we used nine items, employing a 5-point Likert scale (1 = strongly disagree, 5 = strongly agree). The items included were: (1) I feel oppressed and exhausted. (2) I feel tense and worried. (3) I feel calm. (4) I am very worried about falling ill. (5) I am very concerned about family members falling ill. (6) I am very concerned about friends falling ill. (7) I am very afraid of falling ill. (8) I am very afraid of family members falling ill. (9) I am very afraid of friends falling ill. By averaging the responses, we constructed a negative emotion index (M = 3.43, SD = 0.81, Cronbach’s α = 0.91).

#### 2.2.3. Information Seeking

Referring to the study by Lee and Cho [51], we specifically asked respondents the following: (1) I will proactively search for relevant information soon. (2) I will discuss relevant information with doctors, family, and friends. (3) I wish to acquire more information about such diseases. Measurements were made using a five-point scale (1 = strongly disagree, 5 = strongly agree). The “Information Seeking” index was obtained by adding the scores of the aforementioned items and then taking the average (M = 3.54, SD = 0.93, Cronbach’s α = 0.82).

#### 2.2.4. Health Consumption Behavior

This study concentrates on proactive expenditure in health consumption behavior, specifically, the behavior of the youth demographic in voluntarily purchasing health products and services. Respondents were specifically asked whether they agreed with the following statements: (1) I intend to purchase health and wellness items (such as health pots, massage devices, etc.). (2) I plan to purchase health insurance. (3) I intend to undergo relevant medical examinations or receive relevant vaccinations. (4) I can purchase gym memberships or other fitness services. Measurements were conducted using a five-point scale (1 = strongly disagree, 5 = strongly agree). The “Health Consumption Behavior” index was obtained by summing up the scores of the aforementioned items and then taking the average (M = 3.64, SD = 0.86, Cronbach’s α = 0.83).

### 2.3. Data Analysis

In the initial analysis, we conducted descriptive statistical analysis on the principal variables using SPSS 26.0 and performed bivariate correlation analyses between them (Table 1). We applied structural equation modeling via AMOS 21.0 to validate the hypothesized relationships in the proposed model. Maximum Likelihood Estimation was utilized to determine coefficients and evaluate the significance of the hypothesized relationships. The model’s adequacy was assessed against the criteria—RMSEA ≤ 0.06, CFI ≥ 0.95, and SRMR ≤ 0.05—deeming the model to be well fitted [52].

In our research, we utilized a structural equation model. By formulating exogenous latent variables for health risk perception and endogenous latent variables for health information seeking and negative emotions, a path analysis was undertaken to discern the determinants of health consumption behavior among the youth demographic. The fit index gauges the congruence between the model and collected data. For the current model, the chi-square to degrees of freedom ratio (X2/df) stands at 2.55, which is <3. The indices of NFI, AGFI, and CFI register at 0.95, 0.93, and 0.97, respectively, denoting a commendable model fit. Additionally, the RMSEA is evaluated at 0.04, falling below the 0.06 benchmark. Given these metrics, the model’s adjustment parameters align with established reference values, signifying a robust alignment of our structural equation model with the dataset. Consequently, the model’s fit was deemed satisfactory, obviating the need for further refinements and establishing its suitability for hypothesis testing.

## 3. Results

### 3.1. Hypothesis Testing for Direct Effects

In our structural equation model analysis, demographic factors like gender and major were integrated as control variables. The results from the direct effect test (Table 2) indicate that health risk perception exerts a significant positive influence on negative emotions (β = 0.77, *p* < 0.001), information seeking (β = 0.49, *p* < 0.001), and health consumption behavior (β = 0.16, *p* < 0.01). This evidence lends credence to Hypothesis 1.

Both negative emotions (β = 0.22, *p* < 0.01) and information seeking (β = 0.47, *p* < 0.001) exhibit a significant positive influence on health consumption behavior. Notably, the standardized path coefficient for information seeking, at 0.49, surpasses the standardized coefficients delineating the relationships between health risk perception, negative emotions, and health consumption behavior. Health consumption behavior in the youth demographic is shaped by health risk perception and the interplay of negative emotions and information seeking. Additionally, negative emotions and information seeking serve as mediating variables linking health risk perception and health consumption behavior among youths.

Furthermore, the significant positive impact of negative emotions on information seeking (β = 0.67, *p* < 0.001) hints at a chained mediation process, wherein the two mediating variables are interconnected in bridging health risk perception and health consumption behavior (Figure 2).

### 3.2. Single Mediator Effect Test

To ascertain the mediating effects of information seeking and negative emotions on the relationship between health risk perception and health consumption behavior, we employed the Bootstrap method. This method necessitates an initial estimation of the standard error and unstandardized coefficients for the mediation effect, followed by calculating the associated confidence intervals and significance levels (Z-values). Utilizing the Bootstrap procedure, we conducted resampling 5000 times, operating within a 95% confidence interval. A significant mediation effect is indicated if the confidence interval does not encompass 0 (bounded by the upper and lower limits). Detailed standardized estimates alongside their standard errors are presented in Table 3.

The data demonstrate that both negative emotions (β = 0.13, Z = 2.72, 95% CI= [0.03, 0.22) and information seeking (β = 0.18, Z = 4.32, 95% CI= [0.11, 0.28]) play significant mediating roles in the relationship between health risk perception and health consumption behavior. With the validation of Hypothesis 1, which posits that health risk perception positively influences health consumption behavior, the mediating roles of negative emotions and information seeking can be understood as partial mediation. The findings indicate that the health risk perception of the youth demographic can influence their health consumption behavior directly and can also exert an indirect influence by amplifying negative emotions or enhancing the propensity to seek information. Both Hypothesis 2 and Hypothesis 3 are substantiated by these findings.

### 3.3. Chained Mediation Effect Test

The examination of multiple mediating effects by Amos can be implemented by creating corresponding codes in the syntax column. Therefore, to verify the mediating effects of negative emotions and information search between health risk perception and health consumption behavior, this study used the Bootstrap procedure and parameter described above to test the significance of the chained mediation effect by creating syntax codes. Table 3 illustrates that health risk perception shows a chained mediation effect on health consumption behavior through negative emotions and information searching, accounting for 30.21% of the total effect. In other words, the effect of health risk perception on the health consumption behavior of the youth demographic is realized not merely through individual mediation of negative emotions or information seeking but also the enhancing effect of negative emotions on information seeking. This represents a classic chained mediation model (β = 0.19, Z = 4.68, 95% CI = [0.12, 0.28]). Thus, Hypothesis 4 is validated by these findings.

## 4. Discussion

Since introducing the HBM and PMT, scholars have delved into understanding the intricate dynamics between individuals’ health risk perception and their health behaviors. Drawing from a comprehensive nationwide survey in China, this study endeavors to expand the discourse on health behaviors into the consumption realm. It scrutinizes how health risk perception among youths influences their health consumption behavior, further elucidating the chained mediation roles of information seeking and negative emotions. The results indicate that health risk perception positively affects health consumption behavior, underpinned by the intertwined mediation of information seeking and negative emotions.

In recent times, young people are increasingly engaging in health consumption behavior, such as purchasing health products, securing vaccinations, and enrolling in gym memberships to counter health risks. This is attributable to the efficiency of health consumption behavior over traditional methods like diet and exercise. These behaviors not only mitigate negative emotions and diminish health risks, maintaining wellness, but also significantly cut the time costs for health-preserving actions among youths. Yet, the existing body of research mainly revolves around traditional health behaviors like quitting smoking [53,54], reducing alcohol consumption [55,56], and maintaining physical activity [57,58]. Consequently, the intricate interplay among health risk perception, negative emotions, information seeking, and health consumption remains under-investigated. This research defines and investigates health consumption behavior, clarifying the role of health risk perception in shaping these behaviors via negative emotions and information searching. It expands the HBM and PMT beyond conventional health preservation actions into consumption realms, thereby fostering interdisciplinary exchanges among health communication, consumption behavior studies, and related disciplines.

This study measures, for the first time, health consumption behavior as a type of healthy behavior and confirmed its positive correlation with health risk perception. That is to say, the higher the degree of one’s health risk perception, the more likely they are to engage in health consumption behaviors. The research not only validated the predictive power of the Health Belief Model and Protection Motivation Theory in the context of China but also revealed an important reason why people are willing to pay for health—to reduce their own potential health risks. For example, in a fast-paced and high-pressure social life, when young people face insomnia issues and consequently perceive a higher level of health risk, they proactively purchase health products like melatonin to aid in sleeping, thus reducing their own potential health risks.

Furthermore, this study establishes that both negative emotions and information seeking act as mediating variables in the process by which health risk perception steers health consumption behavior. On the one hand, individuals’ perceptions of health risks can indirectly trigger the emergence of health consumption behavior by amplifying their own negative emotions of anxiety, worry, and fear. For example, when young individuals see numerous cases of cancer caused by air pollution and consequently feel fear, their likelihood of purchasing masks or air purifiers may increase as a result. This aligns with the PMT, positing that heightened fear strengthens individuals’ motivation to seek protection, leading to the adoption of pertinent protective behaviors to mitigate potential risks [59]. Notably, our results illustrate that beyond just fear, which the PMT corroborates, emotions such as worry and anxiety also emerge as potent psychological catalysts propelling health-protective actions.

On the other hand, this study, by incorporating the variable of information seeking into the structural equation model, provides a clearer elucidation of the impact pathway of health risk perception on health consumption behavior. The research finds that health risk perception can stimulate individuals’ information seeking behavior, thereby increasing their likelihood of engaging in health consumerism. As Griffin et al. [48] pointed out in the Risk Information Seeking and Processing model, when facing specific health risks, individuals tend to conduct more detailed information searches driven by the principle of information sufficiency to evaluate the level of risk threat and explore potential coping strategies. However, with the continuous advancement of information algorithms, individuals in this process may frequently “encounter” consumer products packaged as health protection measures and may purchase them with the purpose of reducing their own health risks. For example, the more times overweight individuals search for the dangers of obesity and weight loss measures online, the higher the likelihood they will be induced to buy weight loss pills.

Lastly, this study finds that negative emotions and information seeking play an intermediary role in the interactive mechanism between health risk perception and health consumption behavior, forming a chained mediation effect. Specifically, negative emotions serve as a precursor to information seeking, meaning individuals’ health risk perception promotes information seeking by generating or amplifying negative emotions, which in turn stimulate the occurrence of health consumption behavior. This process can be described as follows: when individuals face a heightened level of risk perception, they may generate or magnify negative emotions such as anxiety, worry, and fear. These negative emotions prompt them to seek information to alleviate their own feelings of unease and uncertainty, leading to engagement in health consumption behavior while searching for information related to their health risks. For example, when young individuals experience tachycardia due to reasons like staying up late or excessive work pressure, they are prone to developing negative emotions such as anxiety or worry. In order to assess their health risks and alleviate these negative emotions, they usually search for potential threats and coping strategies related to tachycardia on social media, and in this process, make decisions to purchase smartwatches or smart bands—these products not only aid them in reducing their health risks by monitoring their heart rates continuously but are also more convenient and cost-effective compared to visiting a doctor.

The insights garnered from this research provide valuable recommendations for health communication initiatives. First, public health communicators should aim to nurture a more scientific and reasonable perspective on health consumption among youths. This can be achieved by mitigating their negative emotions and enhancing their skills in seeking credible health information. Secondly, considering the notable influence of health risk perception on youth health consumption behavior, such consumption products can serve as effective vehicles for health communication. They present an opportunity to bridge the gap between complex medical research and its understanding among the general young populace. As a practical application, incorporating youth-centric health information on health product packaging can be an innovative step towards this goal.

While this study offers notable insights, there are several limitations to consider. First, as the research relies on cross-sectional data, reciprocal causality may exist. Even though our results indicate clear correlations among health risk perception, information seeking, negative emotions, and health consumption behavior, causality cannot be established from these correlations alone. Future studies could utilize experimental or longitudinal methods to better validate the causal relationships, possibly complemented by panel data. Second, our research on the measurement of health consumption behavior is an operational exploration based on existing experience, and the measurement results may have certain biases. Future research can theoretically explore health consumption behavior from multiple dimensions and levels. Lastly, the sample used in this study may be lacking in representativeness, which may have influenced the accuracy and reliability of the study. Specifically, the respondents in this study came from China, and we lacked data pertaining to other countries; future research can fill this gap.

## 5. Conclusions

This study pioneered the examination of consumption behavior as a facet of health behavior, analyzing the nexus between health risk perception and health consumption behavior among youths via structural equation modeling. The findings indicate that health risk perception significantly fosters health consumption behavior. Within this dynamic, both negative emotions and information seeking play individual mediating roles and collectively constitute a chained mediation effect: health risk perception catalyzes information search through the arousal of negative emotions, thereby encouraging health consumption practices. By elucidating the mechanism underpinning youths’ health consumption behavior, this research offers insights for rectifying irrational health consumption tendencies within this demographic.

## Figures and Tables

**Figure 1 behavsci-14-00879-f001:**
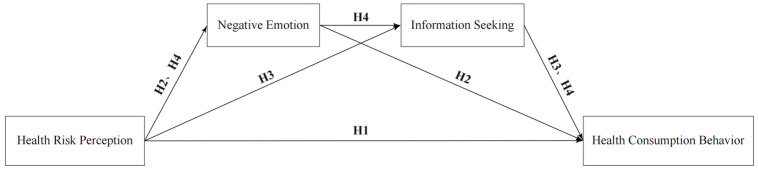
Mechanism Model of the Impact on Health Consumption Behavior. This research aims to analyze the influencing mechanism of health consumption behavior, taking health consumption behavior as the dependent variable and health risk perception as the independent variable. Negative emotion and information seeking are taken as mediating variables. Four research hypotheses (H1–H4) are proposed, and the structural equation model is constructed for verification.

**Figure 2 behavsci-14-00879-f002:**
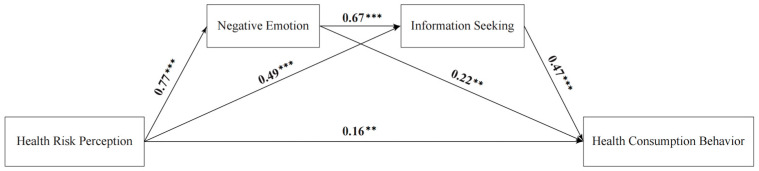
The chained mediation role of negative emotion and information seeking in the relationship between health risk perception and health consumption behavior. Note: ** *p* < 0.01, *** *p* < 0.001.

**Table 1 behavsci-14-00879-t001:** Descriptive statistics and bivariate correlations between key variable.

	Mean	SD	1.	2.	3.	4.
1. Health Risk Perception	3.38	0.87	0.76			
2. Information Seeking	3.54	0.93	0.62 **	0.86		
3. Negative Emotion	3.43	0.88	0.62 **	0.69 **	0.82	
4. Health Consumption Behavior	3.63	0.86	0.53 **	0.60 **	0.58 **	0.81

Note: The numbers on the diagonal represent the square root of the AVE (Average Variance Extracted) of the latent variables, while the numbers below the diagonal represent the correlation coefficients between the latent variables; ** *p* < 0.01.

**Table 2 behavsci-14-00879-t002:** Results of path coefficient estimation modifications and reasons.

			Std. Estimate	S.E.	C.R.	*p*	Unstd. Estimate
Health Risk Perception	--->	Negative Emotions	0.70	0.06	13.02	<0.001	0.76
Health Risk Perception	--->	Information Seeking	0.36	0.06	7.94	<0.001	0.49
Health Risk Perception	--->	Health Consumption Behavior	0.13	0.07	2.36	<0.01	0.16
Negative Emotions	--->	Information Seeking	0.55	0.06	11.97	<0.001	0.67
Negative Emotions	--->	Health Consumption Behavior	0.18	0.07	3.02	<0.01	0.22
Information Seeking	--->	Health Consumption Behavior	0.49	0.07	6.48	<0.001	0.47

**Table 3 behavsci-14-00879-t003:** Report on unstandardized coefficients of mediation effects.

				Bootstrapping				
	Point Estimates	Product of Coefficients		Percentile 95% CI		Bias-Corrected 95% CI		Two- Tailed Sign
		SE	Z	Lower	Upper	Lower	Upper	
HRP--->NE --->HCB	0.13	0.05	2.72	0.03	0.22	0.03	0.22	<0.05
HRP--->IS --->HCB	0.18	0.04	4.32	0.11	0.27	0.11	0.28	<0.001
HRP--->NE--->IS--->HCB	0.19	0.04	4.68	0.12	0.27	0.12	0.28	<0.001

Note: HRP: Health risk perception, NE: Negative emotions, IS: Information seeking, HCB: Health consumption behavior. Bootstrap = 5000.

## Data Availability

The data presented in this study are available on request from the corresponding author.

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
