# Peer review of "Why Are Young People Willing to Pay for Health? Chained Mediation Effect of Negative Emotions and Information Seeking on Health Risk Perception and Health Consumption Behavior"

_behavsci, 2024, doi:10.3390/bs14100879_

Round 1

Reviewer 1 Report

Comments and Suggestions for Authors

Introduction.

The introduction is excessively long. There is information that even being illustrative is not necessary for theoretical understanding. It is not an article of divulgation. My recommendation is eliminated the first paragraph and information since line 170 to 172. Also try to synthesize the information and only let the most relevant theoretical information for the study objective. There are many theories or theoretical models proposed especially as on section 2.3 and 2.4. This can be confusing.

Materials and Methods

Figure 1. Improve the quality of the image.

Again, there is complementary or introductory information that is not necessary as line 267 to 274. The instruments are describing below. So that section is not necessary.

There is a different use in the decimals of the values of the fit index and the values of the correlations. If the journal allows the use of three decimal places, it is fine, but it is necessary to normalize to three decimal places for each value. If the journal does not allow it, you should use only two decimal places as recommended.

Table 2 and 3. Is better put de p values in the table and not the asterisk.

You are mixing results with discussion, as in lines 361 to 365. You need to differentiate between results and discussion analysis.

Table 3. The subtitles of the values are not clear. You can minimize the lyric or improve the format and style of the table.

For our better knowledge would be important have a new figure with the visual results and values of the chained mediation.

Discussion.

Here is where you can be more illustrative and exampling how these relations are functioning. In what cases the young people consume what products for health. After reading the work is not clear for me what healthy behaviors are measuring. Can you clarifying this in the work?.

Comments on the Quality of English Language

The authors need to improve the english. 

Author Response

Comments 1: Introduction. The introduction is excessively long. There is information that even being illustrative is not necessary for theoretical understanding. It is not an article of divulgation. My recommendation is eliminated the first paragraph and information since line 170 to 172. Also try to synthesize the information and only let the most relevant theoretical information for the study objective. There are many theories or theoretical models proposed especially as on section 2.3 and 2.4. This can be confusing.

Response 1: Thank you very much for your suggestions. According to your advice, we have trimmed the introduction. At the same time, we have merged and condensed section 1 and section 2 of the manuscript to better highlight the main points of the article (see lines 26-188).

Comments 2: Figure 1. Improve the quality of the image.

Response 2: Thank you for your valuable suggestions. We have redrawn Figure 1 (see lines 183-188), and to present the research results more clearly, we have added a new Figure 2 to the article (see lines 304-306).

Comments 3: Again, there is complementary or introductory information that is not necessary as line 267 to 274. The instruments are describing below. So that section is not necessary.

Response 3: Thank you for your suggestions. We have deleted and condensed the redundant descriptions in the introduction section (see lines 26-188). Additionally, the text has been polished.

Comments 4: There is a different use in the decimals of the values of the fit index and the values of the correlations. If the journal allows the use of three decimal places, it is fine, but it is necessary to normalize to three decimal places for each value. If the journal does not allow it, you should use only two decimal places as recommended.

Response 4: Thank you for your suggestions. We have standardized all decimal numbers to be kept consistent to two decimal places (see lines 245-246, 256-257, 278-279, 289-290, 317-318).

Comments 5: Table 2 and 3. Is better put de p values in the table and not the asterisk.

Response 5: Thank you for your suggestion. Following your advice, the Table has been described using p-values (see lines 289-290, 317-318).

Comments 6: You are mixing results with discussion, as in lines 361 to 365. You need to differentiate between results and discussion analysis.

Response 6: Thank you for your valuable suggestions. We have revised the description of the results section and moved all discussion points to the discussion section (see lines 284-288, 295-299).

Comments 7: Table 3. The subtitles of the values are not clear. You can minimize the lyric or improve the format and style of the table.

Response 7: Thank you for your valuable suggestions. We have modified the format and font of Table 3 in accordance with your advice (see lines 317-319).

Comments 8: For our better knowledge would be important have a new figure with the visual results and values of the chained mediation.

Response 8: Thank you for your valuable suggestions. We have added Figure 2 to the manuscript to visualize the results, in accordance with your advice (see lines 303-306).

Comments 9: Here is where you can be more illustrative and exampling how these relations are functioning. In what cases the young people consume what products for health. After reading the work is not clear for me what healthy behaviors are measuring. Can you clarifying this in the work?

Response 9: Thank you very much for your valuable suggestions. We have rewritten the discussion section of the manuscript. Additionally, following your advice, we have included some examples to make it easier for readers to understand (see lines 316-370, 373-377, 383-414, 431-437).

Reviewer 2 Report

Comments and Suggestions for Authors

Dear Authors,

Thank you very much for your very detailed and up-to-date work. 

Overall, the results are well presented and, with a few small suggestions for improvement, I would support acceptance for publication on the following condition.

Best regards and wishes!

Suggestions:

- Please consider providing the raw data as a supplement; if necessary, also a presentation of the questionnaire

- In addition to the age range, please add information on the age distribution (mean value, standard deviation) (line 264)

- in the discussion, please explicitly state the limitations of the study (e.g. focus on young population of 1 country)

Author Response

Comments 1: Please consider providing the raw data as a supplement; if necessary, also a presentation of the questionnaire

Response 1: Thank you very much for your valuable suggestions. The raw data and questionnaire have been submitted to the system’s attachments.

Comments 2: In addition to the age range, please add information on the age distribution (mean value, standard deviation) (line 264)

Response 2: Thank you very much for your suggestions. The relevant information regarding the age distribution has been rephrased according to your advice (see lines 197-202).

Comments 3: in the discussion, please explicitly state the limitations of the study (e.g. focus on young population of 1 country)

Response 3: Thank you very much for your valuable suggestions. We have rewritten the discussion section, and the limitations of the study have been clearly stated (see lines 425-437).